# *Quercus suber* Roots Activate Antioxidant and Membrane Protective Processes in Response to High Salinity

**DOI:** 10.3390/plants11040557

**Published:** 2022-02-19

**Authors:** Maria Celeste Dias, Conceição Santos, Márcia Araújo, Pedro M. Barros, Margarida Oliveira, José Miguel P. Ferreira de Oliveira

**Affiliations:** 1Centre for Functional Ecology, Department of Life Sciences, University of Coimbra, Calçada Martim de Freitas, 3000-456 Coimbra, Portugal; celeste.dias@uc.pt (M.C.D.); marciaaraujo@fc.up.pt (M.A.); 2LAQV, REQUIMTE, Faculty of Sciences, University of Porto, Rua do Campo Alegre, 4169-007 Porto, Portugal; csantos@fc.up.pt; 3IB2 Laboratory, Department of Biology, Faculty of Sciences, University of Porto, Rua do Campo Alegre, 4169-007 Porto, Portugal; 4Instituto de Tecnologia Química e Biológica António Xavier, Universidade Nova de Lisboa, Genomics of Plant Stress, Av. da República, 2780-157 Oeiras, Portugal; pbarros@itqb.unl.pt (P.M.B.); mmolive@itqb.unl.pt (M.O.); 5LAQV, REQUIMTE, Laboratory of Applied Chemistry, Department of Chemical Sciences, Faculty of Pharmacy, University of Porto, 4050-313 Porto, Portugal

**Keywords:** salinization, oxidative stress, membrane protection, AP2/ERF family transcription factors, zinc finger CCCH domain-containing proteins, dehydrins

## Abstract

Cork oak (*Quercus suber*) is a species native to Mediterranean areas and its adaptation to the increasingly prevalent abiotic stresses, such as soil salinization, remain unknown. In sequence with recent studies on salt stress response in the leaf, it is fundamental to uncover the plasticity of roots directly exposed to high salinity to better understand how *Q. suber* copes with salt stress. In the present study we aimed to unveil the antioxidants and key-genes involved in the stress-responses (early vs. later responses) of *Q. suber* roots exposed to high salinity. Two-month-old *Q. suber* plants were watered with 300 mM NaCl solution and enzymatic and non-enzymatic antioxidants, lipid peroxidation and the relative expression of genes related to stress response were analysed 8 h and 6 days after salt treatment. After an 8 h of exposure, roots activated the expression of *QsLTI30* and *QsFAD7* genes involved in stress membrane protection, and *QsRAV1* and *QsCZF1* genes involved in tolerance and adaptation. As a result of the continued salinity stress (6 days), lipid peroxidation increased, which was associated with an upregulation of *QsLTI30* gene. Moreover, other protective mechanisms were activated, such as the upregulation of genes related to antioxidant status, *QsCSD1* and *QsAPX2*, and the increase of the antioxidant enzyme activities of superoxide dismutase, catalase, and ascorbate peroxidase, concomitantly with total antioxidant activity and phenols. These data suggest a response dependent on the time of salinity exposure, leading *Q. suber* roots to adopt protective complementary strategies to deal with salt stress.

## 1. Introduction

Soil salinization is a natural and anthropic process that is increasing in the western Mediterranean areas, mostly due to the ecological conditions of this region [1]. In particular, the increasing use of irrigation practices in agricultural lands with poor drainage and high evapotranspiration conditions have also contributed to salt accumulation and acceleration of land degradation in semiarid Mediterranean regions [1]. Therefore, soil salinity is considered an ongoing threat to plant growth in this region and may threaten the native forest ecosystems [2].

Soil salinity is a major abiotic factor contributing to decreased plant productivity and growth, and its occurrence leads to early signalling events, induction of a quiescent phase, and a recovery phase as a result of acclimation responses [3]. Salt stress can induce oxidative stress due to the excessive production of reactive oxygen species (ROS), such as O_2_^−^, H_2_O_2_, and OH [4]. These ROS can damage lipids, proteins, and DNA, resulting for instance, in membrane damages and enzyme activity inhibition [4]. However, at lower levels, ROS also act as signalling molecules mediating salt tolerance [5]. ROS homeostasis during salinity stress is organelle-dependent and relies on the regulation of non-enzymatic (e.g., polyphenols, glutathione, and ascorbate) and enzymatic (e.g., superoxide dismutase, catalase, and ascorbate peroxidase) antioxidant levels, together with metabolic adaptations [5,6].

The cellular and molecular mechanisms involved in adaptation to salinity is starting to be unveiled in model tree species, such as Populus and crops [6,7,8,9]. Several studies support an increase of transcripts associated to antioxidant enzymes in response to abiotic stress [3], namely CSD1 and MSD1 (coding for cytosolic and mitochondrial superoxide dismutase), and CAT2 and APX2 coding, respectively, for catalase and ascorbate peroxidase proteins, however, the stress response remains unclear in cork oak. Several transcription factors (TFs) were also found to be involved in abiotic stress responses. For instance, APETALA2/Ethylene-responsive factor (e.g., RAV1 and ERF1) and zinc finger CCCH domain-containing proteins (e.g., CZF1) are induced by saline stress increasing plant tolerance and adaptation [9,10,11,12,13]. Besides TFs, other proteins encoded by stress-responsive genes, such as dehydrins (e.g., LTI30), play a pivotal role in plant defence interacting with membranes phospholipids, proteins, and DNA, thereby protecting them from damage. In addition, fatty acid desaturases (e.g., FAD7) play a role in membrane fluidity [14]. The response of these membrane protective proteins to salt stress remains to be unveiled in cork oak. In a previous work developed by our team in *Q. suber*, several genes were identified as key-stress responsive genes [15]. According to these previous data, we selected the genes *QsCZF1*, *QsERF5*, *QsFAD7*, *QsLTI30,* and *QsRAV*1 for the present study, in order to quantify their role in the root’s response/adaptation to salt stress.

Cork oak (*Quercus suber* L.) is a xerophytic forest species native to western Mediterranean forest ecosystems and is well adapted to the harsh environmental condition of this region, presenting a good tolerance to hot and dry environments and to low fertility soils [15]. Despite its relevant ecological and economic importance, cork oak has been declining in native areas mostly due to inappropriate land-use policies and extreme weather climate change events that have been frequent in the Mediterranean region [16]. To tackle this potential threat, genomic studies as well as transcriptomic analysis in multiple tissues, developmental stages, and physiological conditions have been developed, mostly related to heat and drought [16,17,18,19]. In this respect, Pereira-Leal et al. [16] identified several expressed sequence tags over-represented in a cDNA library from *Q. suber* roots and shoots under drought, salt, and oxidative stress conditions. Despite the results of this study, the response to salinity in *Q. suber*, particularly at the gene expression level, remains unclear. Most of the studies on salt stress tolerance and adaptation mechanism were performed with other economically important tree species, e.g., pine and eucalyptus [20,21,22]. These studies showed a high tolerance to NaCl concentrations closer to 200 mM in irrigation water. Other works conducted in important crop species, e.g., soybean, rice, and wheat, explored higher salt concentrations, 300 mM, and despite some sensitivity these works provide important information for further breeding programs or genetic manipulation [23,24,25].

We have recently demonstrated [26] that leaves of young cork oak plants exposed to 300 mM NaCl showed oxidative damages, photosynthetic impairment, and chlorophyll decreases. However, in the same study, cork oak leaves also showed features of salt-tolerant species, such as increased energy dissipation mechanisms (non-photochemical quenching), accumulated primary metabolites (soluble sugars and starch), and upregulated antioxidant enzymes to cope with the salt stress. Considering that root surfaces are first exposed to soil salinity, the plasticity of this organ can be the key to cope with salinity. Taking in consideration the previous data of leaf responses to high salinity, we hypothesize that cork oak plants may exhibit tolerance to salt stress, by upregulating protective mechanisms, particularly in the roots. In this study we aim to unveil the protective strategies adopted by *Q. suber* roots during salt stress. *Q. suber* plants were exposed to a high salinity episode (300 mM NaCl) and parameters related to oxidative damages, antioxidant system (enzymatic and non-enzymatic), as well as the expression of genes potentially related to saline stress protection and tolerance were analysed 8 h (early response) and 6 days (later response) after a salt application.

## 2. Results

### 2.1. Oxidative Stress Status

In order to elucidate the general status of oxidative stress in Q. suber roots, the levels of malondialdehyde (MDA), a biomarker of lipid peroxidation, were measured. Additionally, the antioxidant responses developed to counteract these negative effects were analysed by quantifying the total antioxidant activity (TAA) and total phenols. Upon watering with 300 mM NaCl, Q. suber roots showed differential responses in TAA, phenols content, and lipid peroxidation (Figure 1A–C respectively). TAA and lipid peroxidation were significantly decreased 8 h after exposure to salt compared to control. However, after 6 days, TAA and lipid peroxidation were significantly increased in salt-stressed roots (compared to control), with TAA increasing ~25% and MDA levels increasing ~23% (*p* < 0.05). Concerning total phenols content, although levels in roots were not found to be significantly different after 8 h, they significantly increased ~35% after 6 days of watering with 300 mM NaCl when compared to the control.

### 2.2. Antioxidant Enzyme-Encoding Genes and -Activities

Following the assessment of oxidative stress status, *Q. suber* was probed for possible adaptations to the underlying oxidative stress originating from salinity. Herewith, the effect of salinity was determined on the gene expression and activity of superoxide dismutase (SOD), ascorbate peroxidase (APX), and catalase (CAT) antioxidant enzymes. The gene expression of the antioxidant enzymes was not significantly affected 8 h after the saline treatment (Figure 2A). However, after 6 days, relative expression was ~4.5-fold increased for *QsCSD1* and ~2.4-fold increased for *QsAPX2*, in NaCl treatment compared to the control. The activities of SOD, CAT, and APX are represented in Figure 2B. The activities of CAT and SOD in *Q. suber* roots increased significantly after 8 h and 6 days of NaCl treatment, when compared to the control. After 8 h NaCl treatment, APX activity in *Q. suber* roots was not significantly different from the control. However, 6 days after NaCl treatment, APX activity was significantly higher than the control.

### 2.3. Relative Expression of Genes Related to Stress Response

With the aim of investigating the role of putative stress-related genes, relative expression was determined for genes related to saline stress protection and tolerance, most notably *QsCZF1*, *QsERF5*, *QsFAD7*, *QsLTI30,* and *QsR**AV1*. After 8 h of saline stress, relative expression increased for all genes, with significant upregulation observed for the transcription-factor encoding genes *QsCZF1* and *QsRAV1*, and membrane protective genes *QsLTI30* and *QsFAD7*, compared to the control (Figure 3). The largest increase in relative expression, ~6.9 fold, 8 h exposure vs. control, was observed for the *Q**sLTI30* and the second largest increase, ~4.9 fold, 8 h exposure vs. control, was for the *QsRAV1*. Moreover, a significant increase in relative expression (8 h exposure) was found for *QsFAD7* (~3.7 fold) and for the *QsCZF1* (~1.9 fold). After 6 days, compared to 8 h exposure, a general decrease was observed in the expression of the selected genes, significant for the *QsRAV1* and *QsLTI30* genes.

### 2.4. Correlations and Principal Component Analysis

The above-described quantitative data was analysed by Pearson correlation and principal component analysis (PCA) with the goal of identifying trends of variation. Concerning oxidative stress related genes, under the experimental conditions tested, there was a significant positive correlation between *QsCSD1* and *QsAPX2* gene expression (correlation index 1.0, *p* < 0.01), as shown (Figure 4A).

Regarding hypothetical stress-related genes, the genes investigated which were significantly upregulated in saline conditions showed significant positive correlations, viz. *QsLTI30* gene together with *QsRAV1* (correlation index 1.0, *p* < 0.05), and *QsCZF1* gene together with *QsFAD7* (correlation index 1.0, *p* < 0.05). The only cluster observed consisted of dehydrin *QsLTI30* and the two TF genes *QsRAV1* and *QsERF5.*

PCA was performed to identify parameters associated with saline stress in *Q. suber* (Figure 4B). Three well-defined groups represent each condition (C, 8 h and 6 days). The parameters under research were associated with 8 h or 6 days saline stress, but not with the control condition. The parameters associated with early response to saline stress were the expression of the candidate stress-related genes *QsERF5*, *QsRAV1,* and *QsLTI30*, as well as the expression of mitochondrially-encoded superoxide dismutase *QsMSD1*. The parameters both associated with early and late response to saline stress were the expression of the candidate stress-related genes *QsCZF1*, *QsFAD7,* and *QsCAT2*, together with SOD and CAT enzyme activities. The parameters associated with response to late saline stress included the expression of *QsCSD1* and *QsAPX2* genes, APX enzyme activity, lipid peroxidation (measured in MDA equivalents), TAA, and phenols.

## 3. Discussion

Soil salinity is a growing threat to agroforest ecosystems in the Mediterranean, and its effects in forest species is far less known than drought stress, even in xerophytic species such as cork oak. Particularly intriguing is the identity of genes involved in regulating early vs. later responses. We demonstrated previously [26] that the application of 300 mM NaCl in young *Q. suber* plants induced leaf physiological responses as decrease of photosynthesis related parameters, pigments, and carbohydrate profiles. In addition, these studies showed antioxidant responses that were also dependent of the extent of salt exposure. Moreover, several leaf adaptation mechanisms to high salinity were also induced in *Q. suber* plants, suggesting a plastic adaptation of this species to cope with salinity. The roots are the first organ affected by soil salinity, therefore, to complement the previous studied in leaves [26], the present work focuses on this organ.

General oxidative status in the roots of *Q. suber* points to an initial adaptation to stress at 8 h exposure, as evidenced by a decrease in TAA and no change in phenol levels. A decrease in malondialdehyde levels from lipid peroxidation was also observed at this earlier time point. This observation after 8 h, but not after 6 days of saline treatment, suggests that at 8 h, the MDA decrease may not be a direct result of antioxidant enzymes battery, but rather result from other faster protective mechanisms, namely involving the membrane protection. We show that the most responsive protective mechanisms after 8 h can be related with synthesis of dehydrin, e.g., upregulation of *QsLTI30* gene and membrane-associated proteins, e.g., upregulation of *Qs**FAD7* gene. This enhanced synthesis can occur concomitantly with the increase in transcription factors such as CZF1 and RAV1, which regulate several defense genes. Interestingly, these effects were more evident after 8 h compared to a longer period of 6 days, the protective role of the antioxidant enzymes (e.g., SOD and APX) being more evident at this later time. An extensive crosstalk is frequently observed in the plant response against different types of abiotic stress. For example, in *A. thaliana*, LTI30 confers freeze-protection and this is due to its membrane interactions in stressed plants, and this characteristic is also crucial for the maintenance of cell membrane integrity during exposure to other types of abiotic stress such as osmotic or salinity stress [27,28]. Besides protection of membrane phospholipids and proteins, in various plant species dehydrins were reported to bind catalytic metal ions, thereby preventing the Fenton reaction [4] that produces the highly reactive hydroxyl radical. Additionally, in *A. thaliana*, there was an overexpression of *AtLTI30* by drought stress and an increased activity of CAT, leading to a decrease of H_2_O_2_ levels and lower membrane oxidative damage [29]. Considering these previous observations, the large increase in *QsLTI30* expression at 8 h exposure suggests that LTI30 protein may play a role in MDA decrease at this time point. The important protective role of LTI30 at this early period is also supported by the PCA (Figure 4). In addition to this response protection, an increase in SOD and CAT activities (together with the expression of mitochondrial Mn-SOD *QsMSD1*, albeit non-significant), suggests that a metabolic readjustment for detoxification of superoxide is initialized. Besides dehydrins stimulation, after 8 h there was an increase in the transcript levels of *QsFAD7.* The gene encoding FAD7 was reported to be upregulated in wound stressed *A. thaliana* and *Portulaca oleracea* plants [30,31]. Additionally, in *Gossypium hirsutum*, a tissue-dependent *GsFAD7* expression was observed, with reported increased expression in roots and decreased expression in leaves of plants exposed to salt stress conditions vs. control [32]. The overexpression of FAD7 in *Nicotiana tabacum* was also related with increased tolerance to cold stress, while antisense suppression of this gene enhanced plants’ sensitivity to salt and drought stress [33]. The putative involvement of other transcription factors, in the early protective response (8 h) to salinity stress in *Q. suber* roots, is highlighted by the upregulation of *QsCZF1* and *QsRAV1* and suported by the PCA (Figure 4). The involvement of these genes in stress responses/tolerance was also reported for other species. In *A. thaliana* subjected to cold stress, CZF1 and RAV1 TFs were upregulated and also co-regulated [34]. Furthermore, in *Q. suber* roots *QsRAV1* was previously found up-regulated but under drought conditions [35]. Overall, the important role of these stress genes at this early stage of salt stress (8 h) is reinforced by the multivariate analysis (Figure 4). Considering these previous observations, the increased expression of *QsCZF1,* and *QsRAV1* could contribute to a specific adaptation of the cork oak roots to the early stages of salt stress.

With the extent of the salt stress (6 days after NaCl treatment) a higher antioxidant response was boosted by the *Q. suber* roots. This response includes an overexpression of *QsCSD1* and *QsAPX2*, together with increased activities of SOD, APX, and CAT. In addition, both the pool of total phenol and the TAA increase after 6 d of NaCl treatment. The *QsLTI30* gene, related to membrane and protein protection, which was overexpressed at the early stage of salt stress (8 h) remained highly expressed compared to the control at 6 days as well. However, these responses were not sufficient to completely prevent oxidative stress, since lipid peroxidation, a stress biomarker, increased at this stage. The increase of the involvement of enzymatic and non-enzymatic antioxidant battery at day 6 in response salt stress was also evidenced in the PCA (Figure 4).

The role of antioxidant enzymes to deal with salt stress is reported in several studies. For instance, in the woody species, *Broussonetia papyrifera*, salt stress (150 mM NaCl for 5 days) did not increase the levels of H_2_O_2_ in roots, possibly due to the action of the enzyme CAT [36]. Interestingly, in this species the SOD and peroxidase (POX) did not have an important role in stress protection. In the roots of *Lycopersicon pennellii* exposed to a lower NaCl dose (100 mM for 14 days), the upregulation of SOD, CAT, APX and monodehydroascorbate reductase (MDHA) was correlated with the reduction of oxidative stress biomarkers, lipid peroxidation, and H_2_O_2_ [37]. In addition, the increase in the activities and gene expression of SOD, CAT, APX, POX, and glutathione reductase (GR) in *Hordeum vulgare* roots five days after a treatment with 200 mM NaCl was correlated with H_2_O_2_ detoxification [38]. The accumulation of secondary metabolites, such as saponins and lignans that have antioxidant capacity, in Medicago sativa and *Medicago arborea* roots exposed to 100 mM NaCl for 10 days increased salt stress tolerance [39]. In *Olea europaea*, long exposure of roots to 75 and 100 mM NaCl induced the accumulation of polyphenols and the increase of TAA, which was related to protection against salinity stress [40].

## 4. Materials and Methods

### 4.1. Reagents and Standards

Ascorbic acid, sodium borate, disodium EDTA (Na_2_EDTA), dithiothreitol (DTT), ethylene glycol-bis(2-aminoethylether)-N,N,N′,N′-tetraacetic acid (EGTA), gallic acid, hydrogen peroxide, lithium chloride, methanol, methionine, nitro-blue tetrazolium chloride (NBT), polyvinylpyrrolidone (PVP), phenylmethylsulfonyl fluoride (PMSF), triton X-100, potassium sulphate, riboflavin, sodium carbonate, sodium chloride, sodium dodecyl sulfate (SDS), sulphuric acid, thiobarbituric acid (TBA), titanium dioxide, and trichloroacetic acid (TCA), Triton X-100 were purchased from Sigma-Aldrich (St. Louis, MO, USA).

### 4.2. Plant Material and Experimental Design

*Q. suber* half-sibling acorns (collected from one open pollinated mother tree growing at Instituto Superior de Agronomia in Lisbon, Portugal) were germinated in plastic dark pots (400 mL) with a soil mixture of turf and vermiculite (2:1) as described in Ferreira de Oliveira et al. [26]. Seedlings were grown in a climatic room with a photoperiod of 16 h, a temperature of 22 ± 2 °C and a photosynthetic photon flux density of 200 ± 10 μmol m^−2^ s^−1^ for two months. After this period, two-month old plants were randomly divided into three groups: group (1) control—plants watered with 75 mL of water and samples collected after 8 h (n = 4) and 6 days (n = 5); group (2) 8 h (8 h) stress exposure—plants watered once with 75 mL of a solution of 300 mM NaCl and samples collected 8 h after this treatment (n = 9); group (3) 6 days stress exposure—plants watered once with 75 mL of a solution of 300 mM NaCl and samples collected 6 days after this treatment (n = 9). At the end of the treatments, root samples were harvested, washed with ultra-pure water to remove the rest of the turf and vermiculite, immediately frozen in liquid nitrogen, and stored at −80 °C for further determination of the relative expression of stress related genes, parameters related to the oxidative stress and antioxidant system.

### 4.3. Total Antioxidant Activity (TAA) and Phenol

Frozen root powder samples were mixed with methanol, sonicated (40 °C, 30 min), and centrifuged (15,000× *g*, 15 min, 4 °C). The supernatant was used for TAA and phenols content determination. TAA was measured using the ABTS (2,20-azino-bis(3-ethylbenzothiazoline-6-sulphonic acid)) scavenging assay [41]. ABTS solution was added to the extract and the absorbance was read at 734 nm. The TAA was calculated from a gallic acid standard curve. The amount of phenols was quantified by incubating for 5 min the Folin–Ciocâlteu reagent with the supernatant and then adding 20% Na_2_CO_3_. After 2.5 h, the absorbance was read at 765 nm and a gallic acid standard curve was used to determine the phenols content.

### 4.4. Lipid Peroxidation

Frozen root powder was used to assess lipid peroxidation by measuring the production of MDA [42]. Samples (100 mg) were mixed with 1.5 mL of 0.1% (*w*/*v*) TCA and centrifuged (10,000× *g*, 10 min, 4 °C). An aliquot per sample (250 μL) was incubated at 95 °C for 30 min with (a) 1 mL of 20% TCA (*w*/*v*) and 0.5% TBA (*w*/*v*) (+TBA, positive control) and, another (same sample); and with (b) 1 mL of 20% TCA (*w*/*v*) (−TBA, negative control). Samples were immediately cooled on ice and centrifuged (10,000× *g*, 10 min, 4 °C). The absorbance was read at 440, 532, and 600 nm using a Thermo Scientific spectrophotometer (Genesys 10-uv S) and the MDA equivalents were calculated according to Hodges et al. (1999) [42].

### 4.5. RNA Extraction, Primer Design and qPCR

Total RNA was extracted from roots following the Hot Borate method [43]. Total RNA was concentrated and purified (RNeasy MinElute Cleanup Kit, QIAGEN, Hilden, Germany). Before cDNA synthesis, total RNA samples were incubated with Turbo DNA-free (Turbo DNA-free Kit, Ambion, Life Technologies, Austin, TX, USA) to remove contaminating genomic DNA. Prior and after this treatment, total RNA was quantified on a Thermo Fisher Scientific spectrophotometer (NanoDrop 1000). After Turbo DNA-free treatment, 1 µg total RNA was used to synthesize cDNA (Transcriptor High Fidelity cDNA, Roche, Switzerland) according to the manufacturer instructions. cDNA was diluted in ultrapure Milli-Q water (1:10). All qPCR reactions were performed using iTaq SYBR Green Supermix (BioRad, Hercules, CA, USA), 150 nM each gene-specific primer, and 3 μL pre-diluted cDNA in a final volume of 12 µL. The reactions were subjected to 1 cycle at 95 °C for 6 s and 60 cycles at 94 °C for 5 s, 58 °C for 15 s, and 72 °C for 15 s with fluorescence reading in the extension step. At least three qPCR technical replicates were performed per sample. A melting program was performed at the end of qPCR. Average PCR efficiency values and cycle thresholds (CTs) were estimated from the fluorescence data and used to calculate the relative gene expression [44]. Candidate stress-related genes were selected from the analysis of publicly available expression datasets obtained for cork oak tissues in response to abiotic (drought, heat and cold, salt) and biotic stress (*P. cinnamomi* infection). These datasets are available at www.corkoakdb.org (last accessed 1 March 2021). Gene-specific primers (Table 1) were designed using Primer 3 design tool [45] with published *Quercus* consensus sequences. Gene expression levels of target genes were calculated relative to control and double normalised with *TUB2* (tubulin) and *ACT7* (actin) reference genes according to the ΔΔC_T_ method [45], after C_T_ transformation with efficiency values.

### 4.6. Antioxidant Enzyme Activities

Root frozen samples powder were homogenized with an extraction buffer that contained 0.1 M potassium phosphate buffer pH 7.5, 0.5 mM Na_2_EDTA, 0.2% Triton X-100 (*v*/*v*), 2 mM DTT, 1 mM PMSF, and 1% PVP (*m*/*v*). After centrifugation (12,000× *g* for 10 min at 4 °C), the supernatant was used to measure the activities of the enzymes superoxide dismutase (SOD, EC1.15.1.1), ascorbate peroxidase (APX, EC 1.11.1.11), and catalase (CAT, EC 1.11.1.6). The activity of SOD was measured according to Agarwal et al. [46]. SOD activity was assessed at 25 °C by following the reduction of the absorbance at 560 nm produced by the inhibition of the decrease of nitroblue tetrazolium chloride. The APX activity was determined using the protocol developed by Nakano and Asada [47]. H_2_O_2_ was used to start the reaction, and the reduction in the ascorbate was followed at 290 nm. CAT activity was measured following the method of Beers and Sizer [48]. The decrease of absorbance at 240 nm at 25 °C was followed after the addition of 20 mM H_2_O_2_.

### 4.7. Data Analysis

SigmaPlot for Windows version 3.1 was used to perform all statistical analysis. A t-test was performed to compare the control against each of the two experimental groups. Pearson’s correlation was performed to evaluate the relationships between control and stress treatments among all parameters. PCA was performed with prcomp function of R software for Windows.

## 5. Conclusions

Our study provides an integrated overview of the early and later responses of *Q. suber* roots to a salt stress episode. The genes previously reported as over-represented in abiotic and biotic stress were investigated and validated in this study for different time points and ascribed to different putative functions. Data suggest that *Q. suber* response to a high dose of NaCl depends on the extent of the salinity exposure (8 h or 6 days). In an early stage (8 h), *Q. suber* upregulate genes (*QsRAV1*, *QsLTI30, QsCZF1,* and *QsFAD7**)* that can provide some salt tolerance and adaptation, with particular emphasis to genes related with the synthesis of dehydrins and/or membrane protection, which might be associated with the observed low levels of lipid peroxidation. Additionally, the antioxidant enzymes, SOD and CAT, also contribute to this early salt stress response, possibly acting in the detoxification of superoxide. Then, with the extent of stress (6 days) and appearance of signs of oxidative damages (increase of lipid peroxidation), besides the upregulation of *QsLTI30*, other genes related to oxidative stress/antioxidant defence system (*QsCSD1* and *QsAPX2*) were activated, together with a boost in the antioxidant enzyme activities (e.g., SOD and APX), TAA, and phenols. In conclusion, *Q. suber* roots’ early responses to high salt stress point to initial mechanisms of membrane protection, dehydrins synthesis and/or transcription factors regulating protective genes, together with antioxidant enzymes activation. However, with the extent of the salinity, the latter responses include complementary antioxidant protective strategies to deal with the increase of oxidative stress. These data confirm the high tolerance of this species to salinity and reveal the strategies activated by roots to cope with this stress condition. Beside the analysed parameters, other important stress biomarkers such as osmolytes (e.g., proline or glycine-betain),nutrient contents, and reactive oxygen species (e.g., H_2_O_2_) deserve further studies to understand their role in *Q. suber* response and tolerance to salt stress.

## Figures and Tables

**Figure 1 plants-11-00557-f001:**
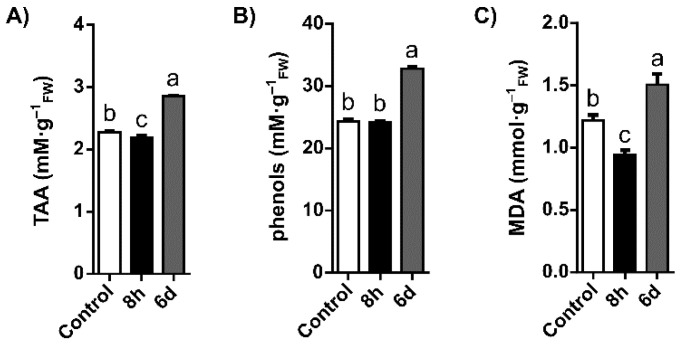
Root total antioxidant activity (**A**), total phenol content (**B**), and malondialdehyde level (**C**) in *Q. suber* roots under control and salinity conditions (8 h and 6 days after 300 mM NaCl treatment). White bars: control; black bars: 8 h; grey bars: 6 days. Values are mean ± SEM (n = 6). Letters indicate significant differences between groups (*p* < 0.05).

**Figure 2 plants-11-00557-f002:**
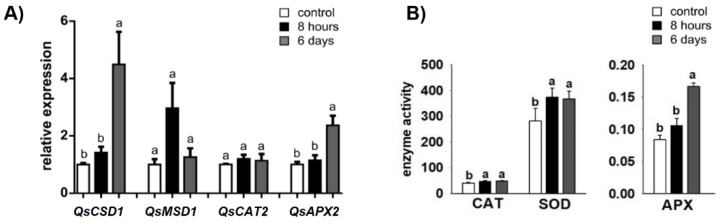
Relative expression levels of genes encoding antioxidant enzymes (**A**) and antioxidant enzyme activities (CAT—catalase, SOD—superoxide dismutase, and APX—ascorbate peroxidase) (**B**) in *Q. suber* roots under control and salinity conditions (8 h and 6 days after 300 mM NaCl treatment). The target genes were: copper/zinc superoxide dismutase (*QsCSD1*), manganese superoxide dismutase (*QsMSD1*), catalase (*QsCAT2*), and ascorbate peroxidase 2 (*QsAPX2*). White bars: control; black bars: 8 h; grey bars: 6 days. Values are mean ± SEM (n = 3–6). For each gene or enzyme different letters indicate significant differences between control and salinity treatments (*p* < 0.05).

**Figure 3 plants-11-00557-f003:**
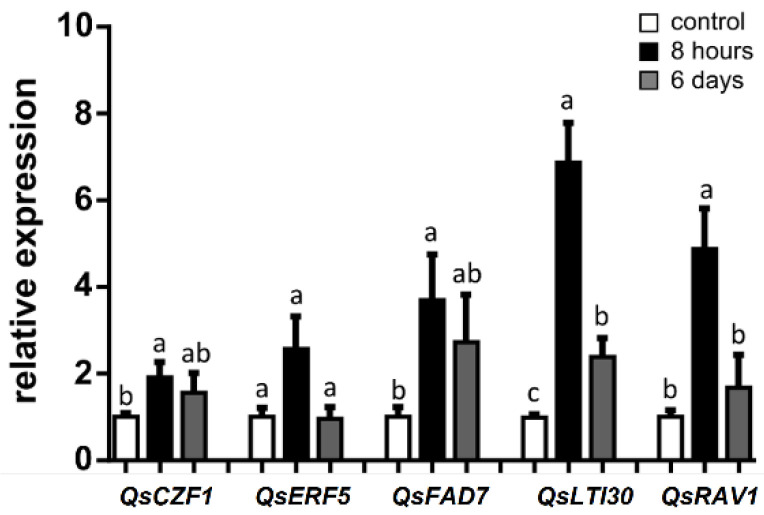
Relative expression levels of genes encoding stress biomarkers in *Q. suber* roots under control and salinity conditions (8 h and 6 days after 300 mM NaCl treatment). Values are mean ± SEM (n = 6). For each gene, different letters indicate significant differences between control and salinity treatments (*p* < 0.05).

**Figure 4 plants-11-00557-f004:**
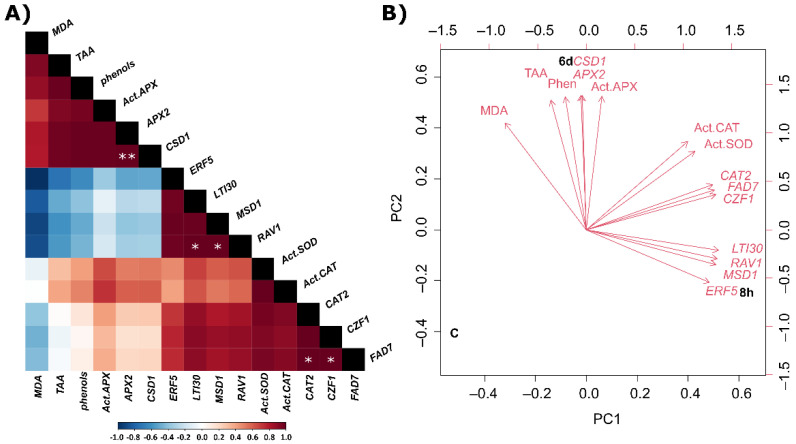
Correlation and PCA of the data. (**A**) Pearson correlation coefficients indicate negative (blue) or positive (red) correlations. * *p* < 0.05; ** *p* < 0.01.; (**B**) PCA biplot of the data in *Q. suber*. C—control, 8 h—8 h after NaCl treatment, and 6 d—6 days after NaCl treatment.

**Table 1 plants-11-00557-t001:** Primers used for qPCR.

Primers (5′—3′)	Target RNA *Q. suber* (NCBI ID)	*A. thaliana* Gene/Locus	Tblastx E-Value (Coverage)
F_CTGAGCGGGAAATTGTTCGTG; R_GCAGTCTCCAACTCCTGCTC	actin (XM_024037498.1)	*ACT7* AT5G09810	0.0 (70%)
F_TACTGTCCCAGAGCTCACCC; R_CACGGAACATAGCTGAGGCA	tubulin (KJ563262.1)	*TUB2* AT5G62690	0.0 (96%)
F_CGCTCTTGGAGACACAACAA; R_CCATCAGCACCAACATTGAC	superoxide dismutase [Cu-Zn] 4A (XM_024050079.1)	*CSD1* AT1G08830	3 × 10^−87^ (59%)
F_CCTTCTCTCTTCCCGATCTCTC; R_GGAGTCGCCTTTAGCGATG	superoxide dismutase [Mn] (XM_024016741.1)	*MSD1* AT3G10920	2 × 10^−124^ (54%)
F_GCTAGGGGAGCTAGTGCAAAG; R_CAGGGTTTCAGGGCTACCA	catalase isozyme 3-like (XM_024064618.1)	*CAT2* AT4G35090	0.0 (81%)
F_CGGATCATCTGAGGGATGTATT; R_CAAAAATAAGAGGGTTGCTGGTC	L-ascorbate peroxidase, cytosolic (XM_024061016.1)	*APX2* AT3G09640	2 × 10^−138^ (61%)
F_GCTCCTTGAGGCATCACACT; R_AGGCGATGGAGTTGGTTCTG	Zn finger CCCH domain-containing protein 29-like (XM_024044287.1)	*CZF1* AT2G40140	0.0 (58%)
F_GCCGTTATCTCCGCATCCTT; R_CCACCAGCACCATTAGCAGA	E-responsive TF ERF105-like (XM_024066622.1)	*ERF5* AT5G47230	4 × 10^−26^ (24%)
F_GACCCACCTCATACAAGCCC; R_TTAGCCCCCAGTTCCTGTCT	omega-3 fatty acid desaturase (XM_024037902.1)	*FAD7* AT3G11170	2 × 10^−135^ (38%)
F_ATATGGCAACCCAACCCACC; R_CATACCTTGGAGAGTGGCGG	dehydrin Xero 1-like (XM_024052398.1)	*LTI30/XERO2* AT3G50970	2 × 10^−13^ (46%)
F_AAGCCGGTTCAACTTTCCCA; R_CGCGTGAACAGCTTTTTGAGA	AP2/ERF and B3 domain-containing TF RAV1-like (XM_024016479.1)	*RAV1* AT1G13260	2 × 10^−102^ (66%)

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
