# Peer review of "Quercus suber Roots Activate Antioxidant and Membrane Protective Processes in Response to High Salinity"

_plants, 2022, doi:10.3390/plants11040557_

Round 1

Reviewer 1 Report

In the current research, the author performed a salt stress experiment and checked the antioxidant and salt-responsive gene expression patterns in the root tissue of the Cork oak plant. From, the analysis author suggested a response dependent on the time of salinity exposure (8 hours or 6 days), leading Cork oak roots to adopt protective complementary strategies to deal with salt stress

The topic is relevant and the study conducted may be useful for the Cork oak research for growing the tolerant plant salt prevent area. However, the writing of MS need to improve substantially there are too many grammatical errors, and very long and confusing statement given need to be polished.

Firstly, results and discussion is not written properly, need to revise extensively. Secondly, can you please explain why the author does not check the difference in root physiology, morphology, or other root traits of control Vs treatment plants?

Apart from this, there are several errors in writing hence before it considers for publication in plants it must go through substantial revision. Hence I would like to endorse for Major revision.

Abstract: ( Need to rewrite) do not emphasize more on previous work, please highlight your current work.

L19-20 “during a high-salinity episode” word episode is not appropriate kindly revise it.

L20-23 Sentence is having a lot of grammatical errors change sentence as below

“Uncovering the plasticity of roots directly exposed to high salinity will also help researchers better understand how this xerophytic forest species copes with salt stress”

“In this work” replace “In the present study…..”

Somewhere written “Hours” “h” please only one type and maintain the consistency throughout the MS. Give space while writing for example “8 h”

L29-34 It's a very lengthy sentence split into two and rewrite for clarity.

For example: “As a result of the continued salinity stress (6 days after salt treatment), lipid peroxidation increased, which was associated with an upregulation of QsLTI30 gene”. Second sentence ………………

Introduction:

L41-42 Please provide a reference.

L59-61 Sentence is long and not clear please revise it.

L91-92 – Revise it not clear “transcripts over-represented in a cDNA library” is obvious what is “(L-18)” why it is written like this.

L102-106 sentence is not appropriate and clear please revise it. For example “Those leaves also showed” what is those leaves can not start a sentence like this.

Results:

All the in result sections mentioned authors need to describe why they did a particular analysis (like antioxidant enzyme-activity, relative expression ………………etc) and then describe the results that these results suggest........

L-120 As the author defined “TAA” in the abstract, it must be defined once again in the main text so please define it and then use the abbreviation TAA. Similarly, follow the other abbreviated terms check throughout the MS. (For example MDA).

L125 “8h” give space “8 h” and use “hour” or “h” consistently

Figure 2. Please add the Bar legends (like Figure 3) in the figure, white control, black 8 h……. it's easy to follow the reader rather than reading the text mentioned below in the figure.

200-207 Please describe the results in the series not starting from LTI30 expression starting from CZF1, also correctly write the gene names in the text as mentioned in the figure “QSCZF1” similarly check others. Also when you are mentioning transcription factor it should not be written in italic, however, the author checked the gene expression so need to make a change in “transcription factor” to “transcript level”.

L225-231 Genes name no need to mention again full name as it defined in the introduction so please write only QsRAV1, QsCZF1.

Relative expression of all the stress biomarkers genes highly expressed during 8 h, and during 6 days, its declined what may be the reason, Secondly, MDA activity was observed highest at 6 days after treatment then it should cause the oxidative damage to plants, then how the plant elevated the salt stress can you please explain?

Figure 4. PCA biplot please put explained variance (%) in the figure. Maybe it is ideal to show 8 h and 6 days separate biplot.

 Discussion: Need the extensive revision

L270-286 Author mentioned the previous studies related to freezing and drought stress, need to show the co-relation with slat stress-related studies.

Most of all the studies mentioned are related to other abiotic stresses and wound stress I do not understand the premise.

 “The QsLTI30 gene, related to membrane and protein protection, that was already overexpressed at the early stage of salt stress (8h) remained overexpressed also to the 6d” This is not true, the author can say it remains significantly higher than control but can't say remain overexpressed.

Author Response

Dear Editor, We would like to acknowledge the valuable comments, remarks, and suggestions made by the reviewers that helped us improve the manuscript's quality. In what follows, we describe the changes on the manuscript and provide a list of corrections/changes made in the manuscript, and we answer the Reviewers’ comments. We also highlighted in manuscript the changes that we did (marked in yellow colour).  

Reviewer 1

Comment: In the current research, the author performed a salt stress experiment and checked the antioxidant and salt-responsive gene expression patterns in the root tissue of the Cork oak plant. From, the analysis author suggested a response dependent on the time of salinity exposure (8 hours or 6 days), leading Cork oak roots to adopt protective complementary strategies to deal with salt stress

The topic is relevant and the study conducted may be useful for the Cork oak research for growing the tolerant plant salt prevent area. However, the writing of MS need to improve substantially there are too many grammatical errors, and very long and confusing statement given need to be polished. Firstly, results and discussion is not written properly, need to revise extensively. Secondly, can you please explain why the author does not check the difference in root physiology, morphology, or other root traits of control Vs treatment plants?

Apart from this, there are several errors in writing hence before it considers for publication in plants it must go through substantial revision. Hence I would like to endorse for Major revision.

Response: We unknowledge the Reviewer comments. Concerning the grammatical errors and the long sentence we improved this part in all the sections of the manuscript, including the Results and Discussion sections. Please see the new version of the manuscript.

As we stated in the manuscript, the main goal of this work was to unveil the antioxidants (enzymatic and non-enzymatic) and key-genes involved in stress-responses (early vs. late responses) of Q. suber roots exposed to high salinity and not mainly focus on other type of analyses (e.g. morphology). We understand that the suggested analyses would provide important information on root salt stress response but we chose to focus on more specific parameters with clear differences

Comment: Abstract: (Need to rewrite) do not emphasize more on previous work, please highlight your current work.

Response: The abstract was changed. Please see the new version of the manuscript.

Comment: L19-20 “during a high-salinity episode” word episode is not appropriate kindly revise it.

Response: The term “episode” in the abstract was deleted and the sentence rewritten. Please see the new version of the abstract.

Comment: L20-23 Sentence is having a lot of grammatical errors change sentence as below “Uncovering the plasticity of roots directly exposed to high salinity will also help researchers better understand how this xerophytic forest species copes with salt stress” “In this work” replace “In the present study…..”

Response: The sentences were corrected. Please see the new version of the abstract.

Comment: Somewhere written “Hours” “h” please only one type and maintain the consistency throughout the MS. Give space while writing for example “8 h”

Response: We follow the Reviewer indication. Please see the new version of the manuscript.

Comment: L29-34 It's a very lengthy sentence split into two and rewrite for clarity. For example: “As a result of the continued salinity stress (6 days after salt treatment), lipid peroxidation increased, which was associated with an upregulation of QsLTI30 gene”. Second sentence ………………

Response: We rewrite two sentences. Please see the new version of the abstract.

Introduction

Comment: L41-42 Please provide a reference. L59-61 Sentence is long and not clear please revise it.

Response: We provided a reference and revised the sentence. Please see the new version of the manuscript (pg. 2, lines 56-57).

Comment: L91-92 – Revise it not clear “transcripts over-represented in a cDNA library” is obvious what is “(L-18)” why it is written like this.

Response: We revised and rewritten the sentence. Please see the new version of the manuscript (pg. 2, line 82).

Comment: L102-106 sentence is not appropriate and clear please revise it. For example “Those leaves also showed” what is those leaves can not start a sentence like this.

Response: We corrected the sentence, please see pg. 2. Lines 96-98.

Results

Comment: All the in result sections mentioned authors need to describe why they did a particular analysis (like antioxidant enzyme-activity, relative expression ………………etc) and then describe the results that these results suggest........

Response: We followed the Review suggestion, and we add some information concerning the analysis performed. Please see pgs.  3, 4 and 5.

Comment: L-120 As the author defined “TAA” in the abstract, it must be defined once again in the main text so please define it and then use the abbreviation TAA. Similarly, follow the other abbreviated terms check throughout the MS. (For example MDA). L125 “8h” give space “8 h” and use “hour” or “h” consistently

Response: We defined the TAA and MDA the first time that we mentioned in the text (please see pg. 3, section Results 2.1.). We introduced a space between 8 and h all over the text.

Comment: Figure 2. Please add the Bar legends (like Figure 3) in the figure, white control, black 8 h……. it's easy to follow the reader rather than reading the text mentioned below in the figure.

Response: We added the bar legend in Fig. 2.

Comment: 200-207 Please describe the results in the series not starting from LTI30 expression starting from CZF1, also correctly write the gene names in the text as mentioned in the figure “QSCZF1” similarly check others. Also when you are mentioning transcription factor it should not be written in italic, however, the author checked the gene expression so need to make a change in “transcription factor” to “transcript level”.

L225-231 Genes name no need to mention again full name as it defined in the introduction so please write only QsRAV1QsCZF1. Relative expression of all the stress biomarkers genes highly expressed during 8 h, and during 6 days, its declined what may be the reason, Secondly, MDA activity was observed highest at 6 days after treatment then it should cause the oxidative damage to plants, then how the plant elevated the salt stress can you please explain?

Response: We changed the description of the results, beginning with the QsCZF1. We also removed unnecessary gene names. Regarding the issue of gene expression, all gene expression data originated from the same biological material, and it is possible that some gene groups show different trends, depending on the function. For example, there is a continuous increase in SOD and APX expression and activity from 8 h to 6 days of exposure. In the same line of reasoning, it is trivial to assume that TF and regulatory genes are activated at early time points (e.g. 8 h) and their regulatory effect is maintained for a longer time period without further requirement for TF upregulation. Regarding the increased MDA levels, they were concomitant with increased SOD and APX activity and expression, as well as increased levels of phenolic compounds, and total antioxidant activity. The most probable conclusion from these two observations is that the elevation in MDA levels observed was not sufficient to impair the root cell responses to oxidative stress, since there was an overall increase in the antioxidant response. Please see the new version of the manuscript.

Comment: Figure 4. PCA biplot please put explained variance (%) in the figure. Maybe it is ideal to show 8 h and 6 days separate biplot.

Response: We have included the variance in PCA plot axes, as requested. For group separation purposes, we opted to maintain the original data so the reader can more easily understand which parameters are closer to the 8 h or to the 6 days conditions. For example, thanks to this visualization, it is possible to see that TF gene expression appears more tightly related to 8 h, while SOD and CAT activity are somewhere in between 8 h and 6 days.

Discussion: Need the extensive revision

Comment: L270-286 Author mentioned the previous studies related to freezing and drought stress, need to show the co-relation with slat stress-related studies. Most of all the studies mentioned are related to other abiotic stresses and wound stress I do not understand the premise.

Response: The studies mentioned in the discussion were more clearly defined according to the objectives and results of our study. In addition, the notion of crosstalk among different types of abiotic stress was made explicit, to evidence the line of reasoning between the presented studies. Please see the new version of the manuscript (pgs. 6 and 7).

Comment: “The QsLTI30 gene, related to membrane and protein protection, that was already overexpressed at the early stage of salt stress (8h) remained overexpressed also to the 6d” This is not true, the author can say it remains significantly higher than control but can't say remain overexpressed.

Response: We corrected the sentence as requested. Please see the new version of the manuscript (pg. 7, line 302)

Reviewer 2 Report

First of all, I want to thank all the authors for their effort in this manuscript

I have to go through the manuscript and find that it was previously reviewed and corrections are being implemented in this version.

I have just some small comments.

1- Abstract is too late and exceeds the limit of 250 words and many details need to be removed in the part of the ur finding.

2- Also the introduction is long and boring in some points regarding the details of the transcription factors of the expressed genes. Try to reduced it.

3- The result section is excellent and all details are clear.

4- the discussion is very good too.

5- the point that the gene expression and antioxidant were not enough to realize the full mechanism of stress tolerance in the plant, it was essential to measure the photosynthetic pigments, osmolytes like proline, and or GB, and finally oxidative stress markers (H2o2 and MDA). Also, the mineral hemostasis was essential to draw how the plant face this stress. My impression was that some more parameters were in need to be measured.

But, the Paper in its form can be passed after adding one or more sentences as a limitation in the conclusion of the paper and recommended some biochemical measurements.

Author Response

Reviewer 2 

We acknowledge the Reviewer’s comments. We follow the suggestions and comments described. Regarding the different points referend by the review:

Comment: First of all, I want to thank all the authors for their effort in this manuscript I have to go through the manuscript and find that it was previously reviewed and corrections are being implemented in this version. I have just some small comments.

1- Abstract is too late and exceeds the limit of 250 words and many details need to be removed in the part of the ur finding.

Response: The abstract was abridged, as requested, to less than 250 words.

2- Also the introduction is long and boring in some points regarding the details of the transcription factors of the expressed genes. Try to reduced it.

Response: We have reduced the introduction by removing unnecessary text repetitions and name of protein effectors not mentioned throughout our study and thus not the most relevant for the discussion.

3- The result section is excellent and all details are clear.

Response: We thank the positive feedback for this section.

4- the discussion is very good too.

Response: We also thank the positive comment for this section.

5- the point that the gene expression and antioxidant were not enough to realize the full mechanism of stress tolerance in the plant, it was essential to measure the photosynthetic pigments, osmolytes like proline, and or GB, and finally oxidative stress markers (H2o2 and MDA). Also, the mineral hemostasis was essential to draw how the plant face this stress. My impression was that some more parameters were in need to be measured. But, the Paper in its form can be passed after adding one or more sentences as a limitation in the conclusion of the paper and recommended some biochemical measurements.

Response: We understand the Reviewer comment. As we stated previously, this work was integrated in a mass-sequencing project that aimed to identify transcripts over-represented in a cDNA library (L-18) from Q. suber roots under drought, salt and oxidative stress conditions, and our specific goal in this work (project task) was to unveil the antioxidants and key-genes involved in stress-responses (early vs. later responses) in roots exposed to high salinity. Indeed, from the proposed mentioned parameters, we only measured MDA levels. The other parameters were unfortunately not measured. However, we added in the final part of the Conclusion section a comment concerning other parameters that can be measured in future studies.

Round 2

Reviewer 1 Report

The authors followed the suggested changes, I would like to endorse the MS for publication. Thank you.

Author Response

Reviewer 1: Comments and Suggestions for Authors: The authors followed the suggested changes, I would like to endorse the MS for publication. Thank you

Replay: We would like to thanks the Reviewer comment.